# *Plasmodium falciparum*-infected erythrocyte co-culture with the monocyte cell line THP-1 does not trigger production of soluble factors reducing brain microvascular barrier function

**Janet Storm** [1], **Grazia Camarda**[1], **Michael J. Haley**[2], **David Brough**[3], **Kevin N. Couper**[2], **Alister G. Craig** [1,4]*

1 Department of Tropical Disease Biology, Liverpool School of Tropical Medicine, Liverpool, United Kingdom, 2 Division of Immunology, Immunity to Infection and Respiratory Medicine, Faculty of Biology, Medicine and Health, University of Manchester, Manchester, United Kingdom, 3 Division of Neuroscience, School of Biological Sciences, Faculty of Biology, Medicine and Health, Manchester Academic Health Science Centre, University of Manchester, Manchester, United Kingdom, 4 Centre for Drugs and Diagnostics, Liverpool School of Tropical Medicine, Liverpool, United Kingdom

* Alister.craig@lstmed.ac.uk

**Data Availability Statement:** All relevant data are within the paper.

## Abstract

Monocytes contribute to the pro-inflammatory immune response during the blood stage of a *Plasmodium falciparum* infection, but their precise role in malaria pathology is not clear. Besides phagocytosis, monocytes are activated by products from *P. falciparum* infected erythrocytes (IE) and one of the activation pathways is potentially the NLR family pyrin domain containing 3 (NLRP3) inflammasome, a multi-protein complex that leads to the production of interleukin (IL)-1β. In cerebral malaria cases, monocytes accumulate at IE sequestration sites in the brain microvascular and the locally produced IL-1β, or other secreted molecules, could contribute to leakage of the blood-brain barrier. To study the activation of monocytes by IE within the brain microvasculature in an *in vitro* model, we co-cultured IT4var14 IE and the monocyte cell line THP-1 for 24 hours and determined whether generated soluble molecules affect barrier function of human brain microvascular endothelial cells, measured by real time trans-endothelial electrical resistance. The medium produced after co-culture did not affect endothelial barrier function and similarly no effect was measured after inducing oxidative stress by adding xanthine oxidase to the co-culture. While IL-1β does decrease barrier function, barely any IL-1β was produced in the co-cultures, indicative of a lack of or incomplete THP-1 activation by IE in this co-culture model.

## Introduction

With malaria cases increasing again after years of effective control, mortality in children is still high with the latest WHO report showing over 600,000 deaths in 2021 [1]. Cerebral malaria (CM) is a syndrome of severe *Plasmodium falciparum* malaria with a ~25% mortality caused by host and parasite interactions. Sequestration of *P. falciparum* infected erythrocytes (IE) in

**Funding:** The study was supported by the Medical Research Council (MR/R010099/1 to KNC, AGC and DB). This was jointly funded by the UK Medical Research Council (MRC) and the UK Department for International Development (DFID) under the MRC/DFID Concordat agreement and was also part of the European and Developing Countries Clinical Trials Partnership 2 programme supported by the European Union. The funders had no role in study design, data collection and analysis, decision to publish, or preparation of the manuscript.

**Competing interests:** The authors have declared that no competing interests exist.

the brain microvasculature and inflammation play a major role in the pathogenesis of CM, leading to leakage of the blood-brain barrier (BBB) and haemorrhages [2].

The innate immune response in a blood stage *P. falciparum* infection engages neutrophils, dendritic cells, natural killer cells and monocytes [3], with the latter playing an important role in contributing to the pro-inflammatory response [4, 5]. Acute malaria is associated with an expansion of monocyte subsets [6], reprogramming of monocytes is seen in children with malaria [7], and an increase in the number of monocytes is observed in controlled human malaria infections in adults [8]. Monocytes also accumulate at IE sequestration sites in the brain of paediatric CM cases [9], potentially contributing to the pathogenesis of CM.

Monocytes (and macrophages) phagocytose IE-derived products, such as extracellular vesicles, *P. falciparum* DNA and haemozoin [3, 10]. The latter can be found in monocytes from patients and more dominantly in cases of severe malaria [4, 11, 12]. Monocytes are also stimulated by pathogen-associated molecular pattern molecules such as DNA-containing immuno-complexes or vesicles [13, 14], haemozoin [15, 16] and glycosylphosphatidylinositol anchors [4]. One of the activation pathways is the NLR family pyrin domain containing 3 (NLRP3) inflammasome, a multi-protein complex that facilitates activation of the protease caspase-1 followed by the maturation and release of the pro-inflammatory cytokines IL-1β and IL-18 from pro-Il-1β and pro-IL-18, respectively [17, 18]. NLRP3 formation requires priming and activation, either with 2 stimuli (canonical) or 1 stimulus (alternative). For canonical NLRP3 inflammasome activation, priming is achieved by damage associated or pathogen associated molecular patterns and activation is initiated by a secondary signal, which causes an acute cellular stress and organelle dysfunction [19]. Alternative NLRP3 inflammasome activation is dependent on only the priming stimulus, which in primary human monocytes serves as both a priming and activation stimulus [20]. Another pathway exists for the sensing of intracellular endotoxin, and this is termed the non-canonical pathway [21]. The NLRP3 inflammasome is activated in the placentas of pregnant women who have malaria [22], and monocytes from malaria patients express NLRP3 inflammasome component proteins [23]. But whether monocyte or macrophage activation by IE-derived soluble factors leads to the inflammasome-dependent production of inflammatory cytokines is debatable with conflicting results from *in vitro* experiments (reviewed in [4, 6]).

Numerous studies have investigated the correlation between levels of inflammatory cytokines in relation to malaria severity with varying results [24]. For IL-1β concentrations in plasma or serum there are contradictory results [25]: no difference was found between clinical malaria syndromes in children [26–29], but differences were reported in adults, with higher levels of IL-1β associated with CM [30] and between mild malaria and SM in patients from all age groups [31]. In these two latter studies, IL-1β was detected in plasma or serum, respectively, at levels of at least 80 pg/ml, while in the other studies most of the IL-1β concentrations were below the detection level of the assays used, except for the study by Lyke et al. [27]. Significantly higher IL-1β concentrations (above detection levels) were found in plasma of CM children that died [28], albeit at relatively low concentrations, and IL-1β expression is induced in the brains of CM cases [32]. This somewhat variable picture suggests an uncertain role for monocytes in malaria pathology, but with the accumulation of monocytes at IE sequestration sites in the brain microvascular, IE-induced generation of IL-1β or other soluble factors could lead to elevated local concentrations. Endothelial cells are also activated by IE-derived products and produce IL-1β [33], and this multi-cellular response may drive blood-brain barrier disruption.

To study the potential role of IE sequestration and monocyte-mediated IE killing within the brain microvasculature, we established an *in vitro* human culture model of IE, monocytes and primary brain microvascular endothelial cells (HBMEC). To avoid variability of monocytes

isolated from donor peripheral blood mononuclear cells (PBMC), the human monocyte THP-1 cell line was used to standardise the contribution of monocytes and they have been widely used to investigate inflammasome activation [34]. For IE, IT4var14 was used, a lab strain that binds to CD36 and intercellular adhesion molecule 1 (ICAM-1) [35]. The model was designed in two stages; firstly, THP-1 cells and IT4var14 IE were co-cultured for 24 hours, then the co-culture medium from this was collected and its effect on brain endothelial barrier function determined by real time trans-endothelial electrical resistance (TEER) [36] and in addition, IL-1β production was measured. As previous work had suggested that oxidative stress seemed to be required to induce IL-1β production, we also performed the co-cultures in the presence of xanthine oxidase (XO), which produces reactive oxygen species from (hypo)xanthine [37].

## Materials and methods

All experiments were conducted at the Liverpool School of Tropical Medicine from January 2019 till October 2021.

### Cell and *Plasmodium falciparum* cultures

THP-1 cells [38] were grown in RPMI 1640 medium supplemented with 10% fetal bovine serum, 2 mM L-glutamine, 100 units/ml penicillin and 0.1 mg/ml streptomycin (RPMI-FBS) and kept between $1x10^5$ - $1x10^6$ cells/ml. Primary human brain microvascular endothelial cells (HBMEC, Cell Systems) were cultured as previously described [36]. Briefly, cells were grown in Endothelial Cell Growth Medium 2 containing 2% foetal calf serum, 5 ng/ml epidermal growth factor, 10 ng/ml basic fibroblast growth factor, 20 ng/ml insulin-like growth factor, 0.5 ng/ml vascular endothelial growth factor 165, 1 μg/ml ascorbic acid, 0.2 μg/ml hydrocortisone and 22.5 μg/ml heparin (EGM2, Promocell), and used between passage 5 and 8. All mammalian cells were grown in a humidified incubator at 37˚C, 5% $CO_2$. The *P. falciparum* lab strain IT4var14 (A4) [35] was cultured in complete RPMI medium (CRM: RPMI 1640 with 25 mM HEPES, 11 mM glucose, 2 mM L-glutamine, 0.2% $NaHCO_3$, 25 μg/l gentamicin, 0.5% Albumax and 0.22 mM hypoxanthine pH 7.4) in normal group O red blood cells (RBC) at 3% haematocrit [36] at 37˚C, in modified atmosphere (1% $O_2$, 3% $CO_2$). Cells and IT4var14 were screened for mycoplasma using MycoBlue™ (Vazyme).

### THP-1/IT4var14 co-cultures

IT4var14 were synchronised with 5% sorbitol at early ring stage and 20–24 hours later trophozoites were enriched by gelatin flotation (0.7% in RPMI medium). 100 x $10^4$ trophozoite stage parasites, at parasitaemias from 50% to 65%, were mixed with 2.5 x $10^4$ THP-1 cells (40:1 ratio) in a Media Mix (MM) composed of 1:1 CRM:RPMI-FBS in a total volume of 150 μl. As controls, THP-1 cells alone, IT4var14 alone, THP-1 plus uninfected RBC (uRBC), and uRBC alone were incubated in parallel conditions. To assess the role of extracellular reactive oxygen species (ROS), Xanthine oxidase (XO, Sigma, Merck) was added to MM at 0.12 lU/ml [37]. Co-cultures were incubated for 24 hrs in a humidified incubator at 37˚C, 5% $CO_2$. At the end of incubation, co-cultures were harvested, centrifuged at 16,000 g for 1min and the collected supernatants were centrifuged again at 16,000 g for 5 min before addition to previously seeded HBMEC. The remaining of these co-culture media were stored at -80˚C. Cellular pellets were smeared on glass slides, fixed with methanol and stained with 10% Giemsa for inspection by microscopy. Images were taken at 1000x magnification.

### Endothelial cell barrier integrity

Barrier function was measured by real time Trans Endothelial Electrical Resistance (TEER) analysis with the xCELLigence® RTCA S16 system (ACEA Biosciences). On the same day of THP-1/IT4var14 co-culturing, HBMEC cells were seeded at 50,000 cells/cm$^2$ (1x10$^4$ cells) in E-Plates 16 PET (ACEA Biosciences) pre-coated with Attachment Factor Protein (Thermo Fisher Scientific), left to settle for 30 min at room temperature, and then transferred to RTCA S16 and recording of the arbitrary cell index (CI) started. The next day, unattached cells were removed by medium change and after CI stabilisation (~2.5 hrs), EGM2 medium was replaced by EGM2 minimal medium (EGM2-min: EGM2 medium lacking heparin and hydrocortisone). After CI stabilisation (~3 hours), half of the medium was discarded and replaced with an equal volume of THP-1/IT4var14 co-culture supernatants, control cell culture supernatants or MM. In negative control wells, half of the medium was removed and re-added to the same well to account for manipulation-induced CI perturbations. TEER signal was recorded over 48 hrs with 4-min intervals. Changes in barrier function were determined after normalisation of the CI at the time point immediately prior to supernatant addition (Normalised Cell Index, NCI) and expressed as % of MM control at each time point (Baseline Normalised Cell Index, BNCI). Barrier function in response to IL-1β treatment was initially performed as above, and after ~3 hours EGM2-min medium, recombinant IL-1β (Biolegend) was added from a 100x concentrated stock solution to final concentrations of 0.01, 0.1, 1, 5 and 10 ng/ml.

### Uric acid and IL-1β measurements

Uric acid and IL-1β levels were determined in (co-)culture supernatants using QuantiChrom™ Uric Acid Assay Kit (BioAssay Systems) and Human IL-1β/IL-1F2 DuoSet ELISA (R&D Systems) respectively, according to manufacturer's instructions. As positive control, THP-1 cells were stimulated with 1 μg/ml lipopolysaccharide (LPS) for a total of 5 hrs, with 10 μM nigericin added during the last hour of incubation.

### Statistical analysis

Data was expressed as mean ± standard deviation (SD) for experiments with less than 5 replicates and as mean ± standard error of the mean (SEM) for experiments with more than 5 replicates. Statistics significance between the culture conditions was calculated by two-way ANOVA using GraphPad Prism 9.5.0. P-values < 0.05 were considered significant.

## Results

### Recombinant IL-1β decreased HBMEC barrier function

Activated monocytes produce IL-1β and it was hypothesised that this cytokine would be produced in the THP-1/IE co-culture. As one of our major readouts was the effect on endothelial barrier function, measured by TEER, the effect of commercially available recombinant IL-1β on HBMEC barrier function was determined; IL-1β decreased the HBMEC barrier function in a dose and time dependent matter (Fig 1). IL-1β, at 100 pg/ml, decreased the barrier function by 15 ± 2% compared to control medium and the maximum effect was reached with a concentration of at least 5 ng/ml after 9 hours, a 25 ± 11% decrease, similar to published data [39].

### THP-1 cells were activated by trophozoite stage IT4var14

For the THP-1/IE co-culture a compromise was made regarding growth medium to keep both the IE and THP-1 viable for 48 hours. A 1:1 mixture of Plasmodium Complete RPMI medium and THP-1 RPMI-FBS medium, named Media Mix (MM), was used. THP-1 cells were

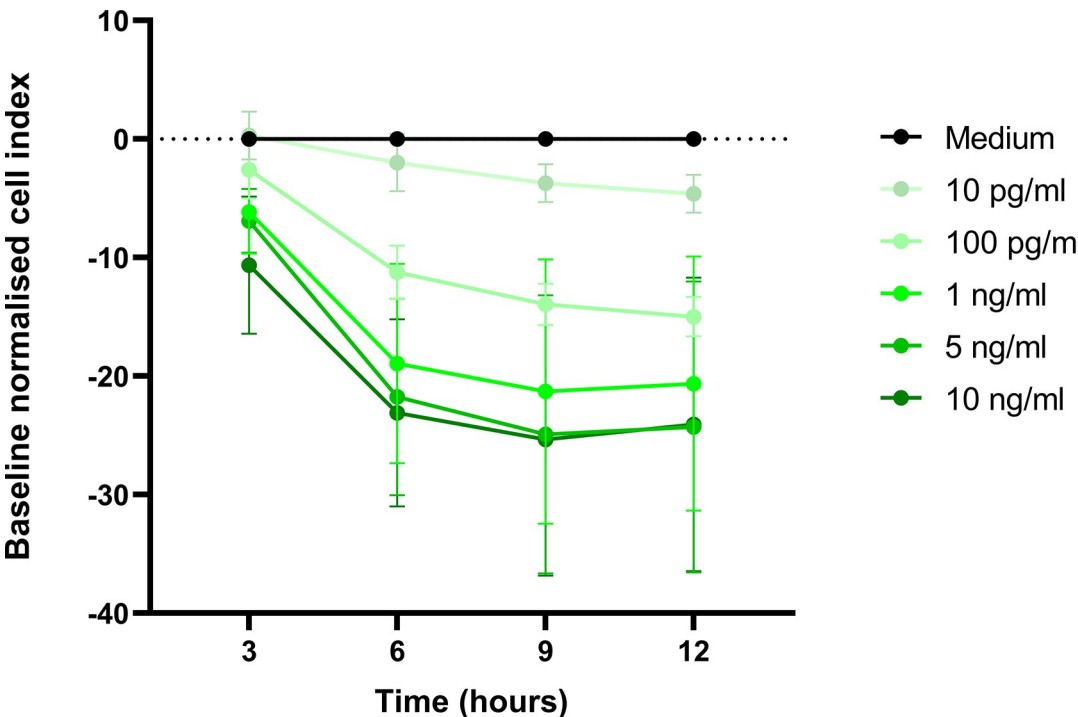

**Fig 1. Dose- and time-dependent effect of IL-1β treatment on HBMEC barrier function using TEER.** Cell index was normalised at the time point immediately prior to the addition of varying concentrations of IL-1β and medium (black line) was set as baseline. Changes in baseline normalised cell index are plotted at the respective time points (mean ± SD of 4 independent experiments).

activated by trophozoite stage IE as shown by the morphological changes occurring after 24 hours: vacuolised cytoplasm, enlarged euchromatic nucleus and nucleoli, and phagocytosis of IE (Fig 2). XO, generating extracellular ROS from (hypo)xanthine in the medium, was added at a concentration of 0.12 U/ml, similar to concentrations found in plasma of malaria patients [40] and used by others [37]. The presence of XO did not appear to affect activation, but it did impair IE development, as previously shown [41] (Fig 2J).

## Co-culture of THP-1 cells and IE did not produce molecules that affected HBMEC barrier function

To assess if the THP-1/IE co-culture produced any soluble factors that affected HBMEC barrier function, co-culture medium was collected after 24 hours and added to a confluent HBMEC monolayer in a TEER E-plate (see Fig 3A for experimental set-up). THP-1/uRBC co-culture medium and 24 hours culture medium of THP-1 cells, IE or uRBC only were used as controls. Barrier function, measured by cell index, was monitored for 48 hours and normalised cell index analysed with MM growth medium as the baseline, shown in a representative plot in Fig 3B. Co-culture of THP-1 cells and IE did not produce factors that affected the HBMEC barrier function compared to the control (co-)culture conditions after 24 or 48 hours (Fig 3C and 3D). The presence of extracellular ROS, by adding XO, did not alter the barrier function either (Fig 3D).

## Co-culture of THP-1 cells and IE produced low levels of IL-1β

That barrier function was not affected by THP-1/IE co-culture medium in the presence of XO was unexpected. Published work showed activation of monocytes in the presence of XO led to

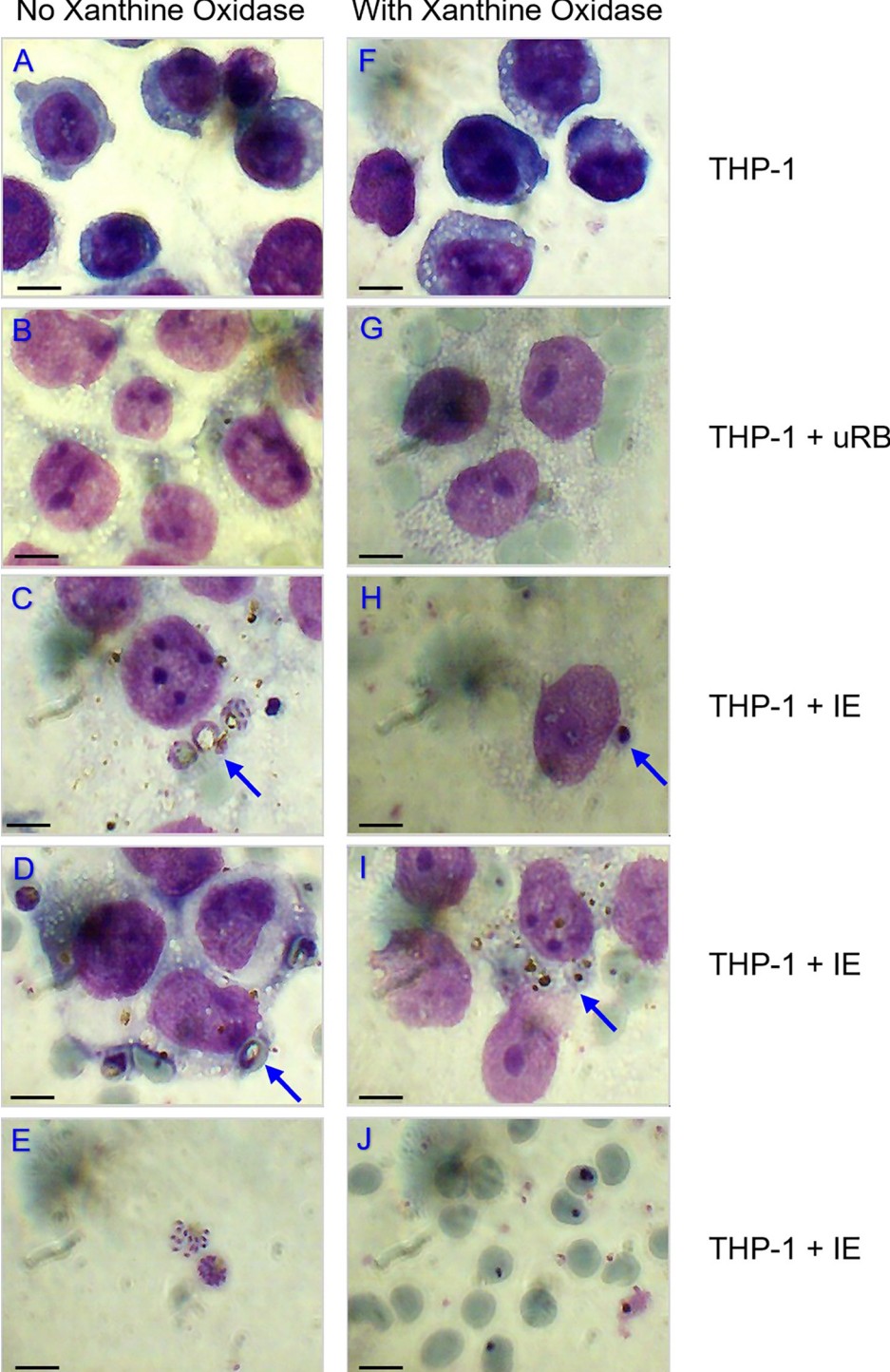

**Fig 2. Morphology of THP-1 and IT4var14 IE after co-culture for 24 hrs.** Images from a representative co-culture experiment in the absence (left, A-E) or presence of xanthine oxidase (right, F-J). A and F: THP-1 cells only; B and G: THP-1 + uRBC; C, D, E, H, I and J: THP-1 + IE. THP-1 cells were activated upon 24 hour co-culture with IE with highly vacuolised cytoplasm, enlarged euchromatic nucleus and nucleoli (C, D, H and I). Phagocytosis of IE occurred (arrows in C, D, H and I) and in co-cultures with xanthine oxidase, parasites did not progress beyond the trophozoite stage and appeared pyknotic (compare J with E). The scale bar is 10 μm.

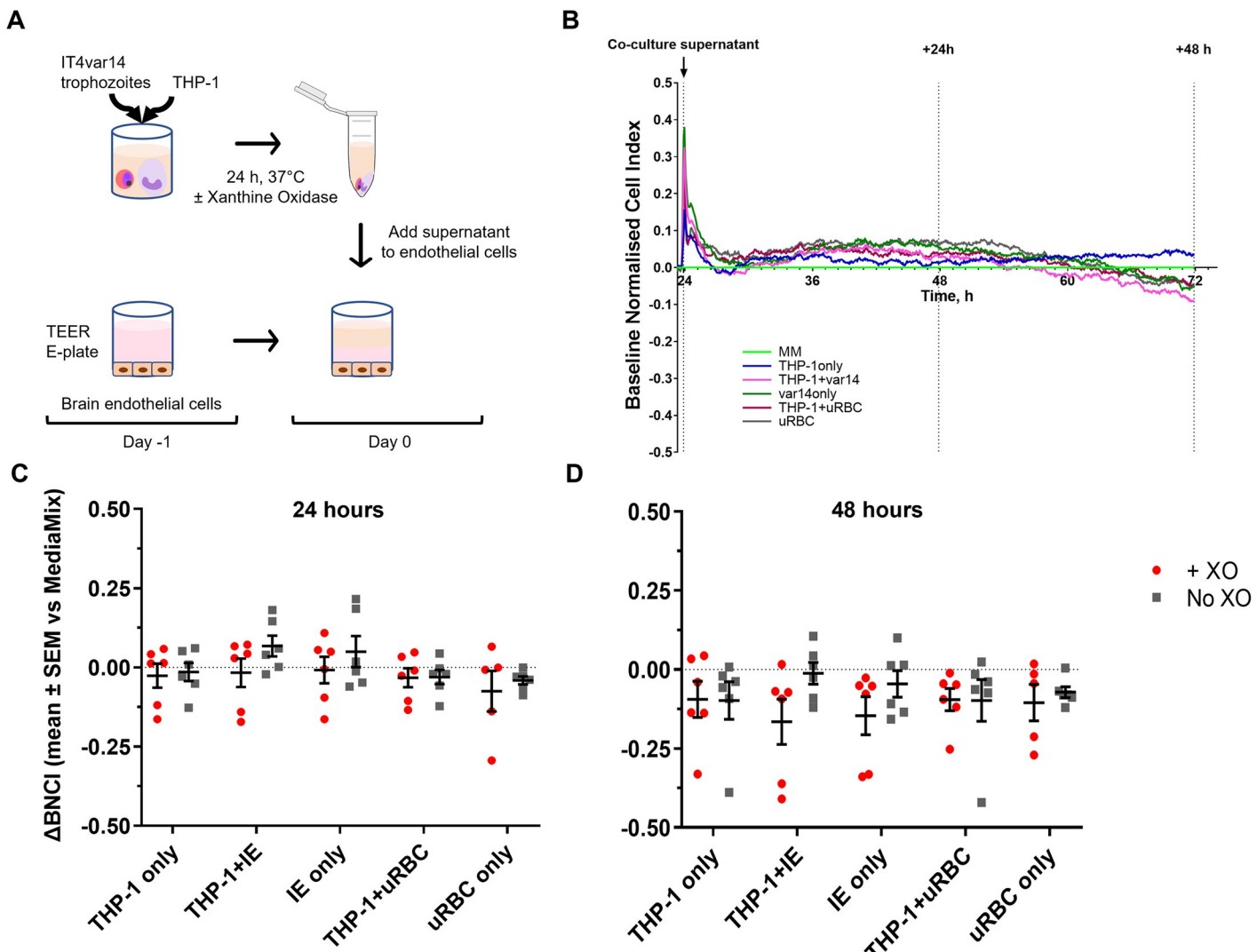

**Fig 3. HBMEC barrier function is not altered by THP-1/IE co-culture supernatants.** (A) Schematic of the co-culture and TEER procedure. (B) Representative TEER trace of HBMEC with the different culture conditions. Cell index was normalised at the time point immediately prior to the addition of the (co-)culture medium (black arrow) with media mix (MM, bright green) set as baseline (baseline normalised cell index, BNCI). (C) ΔBNCI of HBMEC for all the IT4var14/THP-1 (co-)culture conditions in the presence (red dots) or absence (grey squares) of xanthine oxidase (XO) after 24 hours. (D) Same as C with ΔBCNI at 48 hours. Mean ± SEM of 6 independent experiments, no significant difference was detected between the conditions as calculated by two-way ANOVA.

production of IL-1β [37] and our own results showed that concentrations of 10 pg/ml IL-1β produced a significant (but small) decrease in HBMEC barrier function (Fig 1). IL-1β levels were determined in the THP-1/IE and THP-1/uRBC co-culture medium and the THP-1, IE and uRBC control culture media in the presence of XO. None of the conditions generated more than 5 pg/ml, explaining the lack of effect on the barrier function (Fig 4). However, the positive control, THP-1 cells incubated with LPS and nigericin, did produce 13.1 ng/ml IL-1β, showing that the THP-1 cells are capable of producing IL-1β after activation.

## Xanthine oxidase added to culture medium produced uric acid

To examine the activity of XO, its product uric acid was determined in the THP-1/IE co-culture medium. In the presence of XO, uric acid was produced in all the co-culture conditions

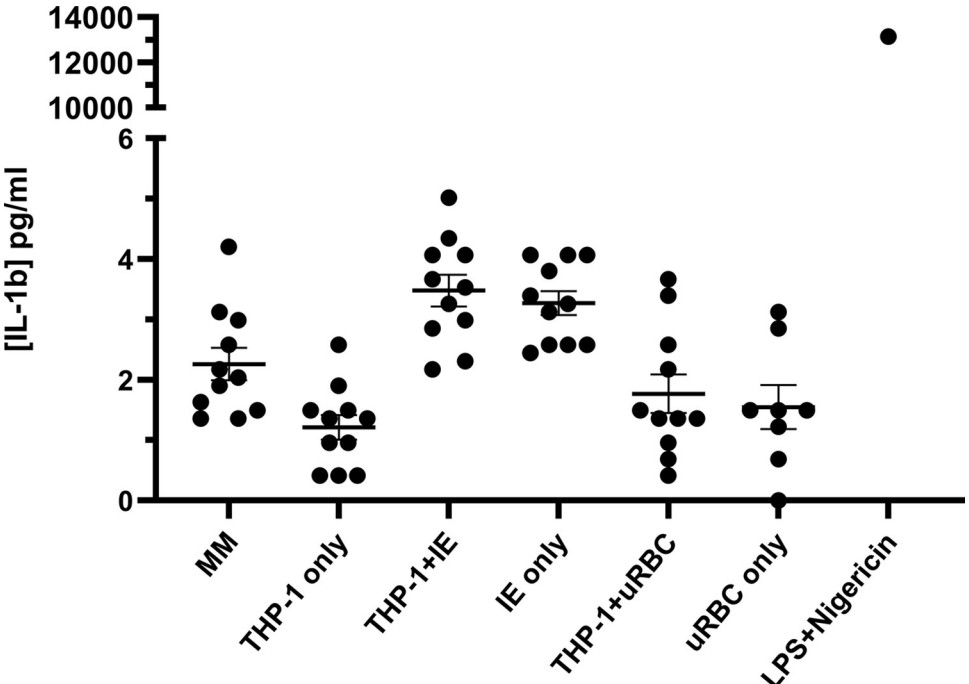

**Fig 4. IL-1β levels in xanthine oxidase treated THP-1/IE co-cultures.** IL-1β levels were measured in 24 hour (co-) culture medium in the presence of xanthine oxidase with LPS + nigericin treatment as control. Mean ± SEM from 11 (8 for uninfected RBC only condition) independent experiments is plotted, of which 9 have also been used in TEER assays. Statistical significance is not shown, as most of the IL-1β concentrations are lower than the sensitivity of the ELISA (3.6 pg/ml).

and controls, including MM growth medium (Fig 5). This growth medium contains hypoxanthine and FBS, thus an abundance of substrate for XO is present. The presence of IE or uRBC increased the amount of uric acid produced to approximately 1.5 mg/dl (15 μg/ml), significantly higher than by THP-1 cells alone. Although uric acid can induce endothelial cell dysfunction [42] and activates the inflammasome in monocytes/macrophages [15], the concentration in the culture supernatants was below normal levels in blood (2–7 mg/dL) and thus unlikely to cause an effect on THP-1 cells or HBMEC barrier function. However, the production of uric acid indicates that ROS were produced by XO in our co-culture conditions.

## Discussion

IE sequestration, inflammation and coagulation responses are main factors in the pathology of CM leading to endothelial dysfunction, disruption of the blood brain barrier and the development of haemorrhages, as summarised in many reviews [2, 43–47]. In recent years, a role for neutrophils and T cells has been identified; an association of activated neutrophils and IE sequestration in Malawian children [48] and a correlation between intravascular CD3+CD8 + T cells and sequestration in post-mortem brain tissue of children [49].

The accumulation of monocytes at IE sequestration sites in post-mortem brain tissue [9] and the activation of monocytes by IE and parasite products led us to investigate whether secreted molecules of IE-activated monocytes affect brain endothelial barrier function. To date only one malaria tripartite model has been published, consisting of the human brain endothelial cell line HBEC5i, PBMC of malaria naïve blood donors and different *P. falciparum* lab strains [50]. The presence of brain endothelial cells amplified the IE-induced production of

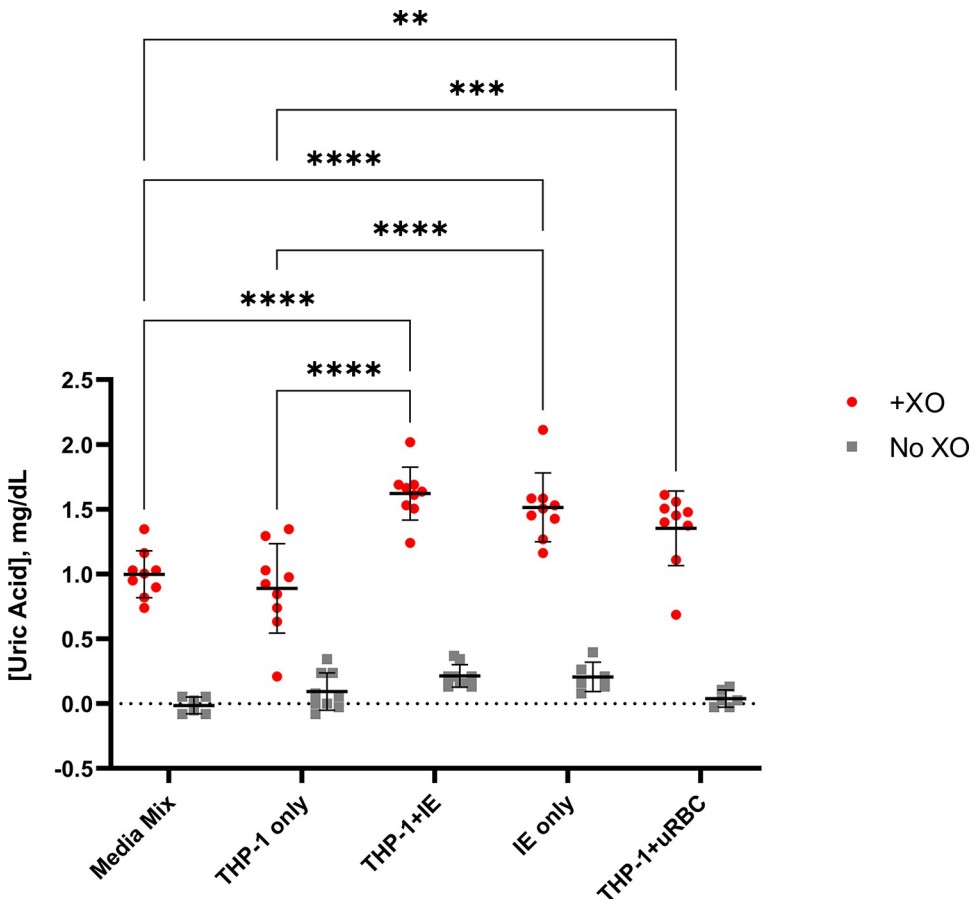

**Fig 5. Uric acid detection in THP-1/IE co-culture medium.** Uric acid levels were measured in supernatants from 24 hour (co-)cultures in the presence or absence of xanthine oxidase (XO). Mean ± SEM from 9 independent experiments is plotted; statistical significance between the +XO conditions was calculated by two-way ANOVA with ** $p < 0.01$, *** $p < 0.001$ and **** $p < 0.0001$.

interferon-gamma by PBMC, with the source likely to be natural killer cells, but no upregulation of IL-1β was seen in the presence of IE compared to the RBC control, or in the presence of brain endothelial cells. The results of this tripartite model were variable for each of the blood donors, indicative of the complexity of tripartite models and variability using primary PBMCs. Our model consisted of primary HBMEC, IT4var14 IE and the monocyte cell line THP-1, representing circulating monocytes that encounter blood stage IE. Monocytes phagocytose both immune serum opsonised and non-opsonised IE and in our THP-1/IE co-cultures non-opsonic phagocytosis of IE was observed (Fig 2). Non-opsonic phagocytosis is dependent on monocyte CD36 expression [51] and IT4var14 IE bind to CD36 [35].

The observation that 24 hours THP-1/IE co-culture did not produce any soluble factors that disrupt the HBMEC barrier function was surprising. Activation of monocytes by IE or by IE products; either whole IE lysate, IE vesicles, DNA-containing immunocomplexes or isolated molecules, such as haemozoin, has been reported in many studies [4, 6, 52, 53]. In particular, ingestion of haemozoin activates the NLRP3 inflammasome leading to IL-1β production in THP-1 cells [15, 16]. As shown in Fig 2E, IT4var14 progresses through their life cycle in 24 hour co-culture in the absence of XO and with an IE:monocyte ratio of 40:1 parasite material is likely to be released. Morphological changes of THP-1 cells indicated that they were

activated by IE (Fig 2B–2D), but this did not result in the production of sufficient quantities of modulators that affect HBMEC barrier function. The concentration of IL-1β generated in the co-culture (<5 pg/ml) was much lower than the 10 pg/ml that had a measurable effect on barrier function as shown in Fig 1.

Perhaps only one signal from IE or IE derived material is not enough to activate the inflammasome in monocytes. Ty et al showed that besides IE lysate, an additional signal was required for the activation of the macrophage inflammasome [37]. Extracellular ROS, which was produced by adding XO to their monocyte-derived macrophage co-cultures, was required to produce IL-1β. However, adding XO to our THP-1/IE co-cultures produced very little IL-1β (< 5 pg/ml), in contrast to the LPS-Nigericin stimulated THP-1 control, which produced significant quantities of IL-1β (~13 ng/ml, Fig 4). Further, co-cultures in the presence of XO did not produce any other molecules that had an effect on HBMEC barrier function (Fig 3). Nevertheless, XO did produce uric acid from (hypo)xanthine in the MM growth medium and the (co)-culture medium, indicative of ROS generation (Fig 5). Uric acid crystals (monosodium urate) also activate the inflammasome at a concentration of 100 μg/ml [15, 16], but the THP-1/IE co-culture with XO only produced ~ 15 μg/ml uric acid, below the effective activation concentration. Higher concentrations of XO could not be used, as concentrations above 0.12 U/ml of solely XO affected HBMEC barrier function.

The lack of IL-1β production in the THP-1/IE co-culture in the presence of XO contrasts with the data on activation of monocyte-derived macrophages by IE and ROS [37], and is likely to be attributed to differences between macrophages and monocytes, and in particular THP-1 cells. While macrophages, differentiated from monocytes isolated from healthy U.S. donors, only produced IL-1β after stimulation with IE lysate combined with XO [37], monocytes, also isolated from healthy U.S. adults, produced IL-1β after stimulation with IE lysate alone [7]. However, in our co-culture model, we used THP-1 cells instead of isolated monocytes and intact IE instead of IE lysate, which could be a reason no IL-1β was produced, even in the presence of XO. The ratio of IE:monocyte used in our co-culture was 40:1, similar or in excess to the above-mentioned studies; an IE lysate:monocyte ratio of 30:1 [7] or an IE lysate: monocyte-derived macrophage ratio of 8:1 [37]. The time of co-culture, 24 hours, was also the same as these two studies. As mentioned, IE-derived molecules are likely to be present after 24 hours co-culture and the THP-1 cells were morphologically activated (Fig 2), but concentrations might not be high enough to trigger NLRP3 inflammasome formation. THP-1 cells were used to avoid variability in PBMC-isolated monocytes and have been widely used to investigate activation of monocytes in *P. falciparum* malaria, but mostly when pre-treated with synthetic phorbol 12-myristate 13-acetate (PMA), which differentiates them into macrophage-like cells [15, 16]. As asexual erythrocytic stage parasites mostly encounter circulating monocytes, we chose not to differentiate the THP-1 cells with PMA and only used IE. Although in a *P. falciparum* infection, monocytes will be activated by circulating and local cytokines and other signals, potentially giving additional priming and activation triggers as well as IE or parasite molecules, we only used XO as stimulus.

From our results it is evident that THP-1 cells behave differently than isolated monocytes in activation of the NLRP3 inflammasome by IE or IE-derived molecules. IL-1β generation by THP-1 cells is achieved by LPS stimulation and further enhanced by the combination of LPS priming and nigericin treatment [54] and as shown in Fig 4, the positive control of LPS and subsequent nigericin treatment of THP-1 cells produced large amounts of IL-1β. The effect of produced IL-1β in this culture supernatant on barrier function was masked by LPS, which in itself decreased barrier function substantially. Unfortunately, we were not able to isolate monocytes from blood donors and compare isolated monocytes with THP-1 cells in our co-culture model. These differences between monocytes, either isolated from PBMCs or the

THP-1 cell line, and monocyte-derived macrophages are often overlooked in reviews of *in vitro* and *ex-vivo* studies into the role of monocytes in malaria [4, 5, 10] and experimental methods have also varied widely as reviewed by Dobbs et al. [6]. Thus, the role of monocytes, and in particular the activation of the NLRP3 inflammasome and Il-1β generation, in *P. falciparum* infection and pathogenesis of malaria is still debatable [55, 56].

Although systemic IL-1β concentrations are low in *P. falciparum* infections, higher concentrations were associated with death in CM cases in Malawian children [28] and local vascular concentrations could be high enough to affect endothelial cell barrier function. Besides production by monocytes, endothelial cells can also produce IL-1β through activation with IE-derived products, such as histidine-rich protein II, [33]. Immunohistochemistry of post-mortem brain sections of CM cases of Ghanian children showed strong intravascular IL-1β staining which was absent in cases without central nervous system infections. The CM cases had also increased ICAM-1 and VCAM-1 expression that correlated with IE sequestration in the microvessels [57]. IL-1β expression was also detected in vessels and infiltrating leukocytes in the brain of Malawian paediatric CM cases [32]. The combination of IE sequestration, monocyte accumulation and endothelial activation and thus local high concentrations of Il-1β in brain microvasculature might contribute to the breakdown of the BBB and haemorrhaging seen in CM.

## Acknowledgments

We thank Jack Green, University of Manchester for help with the IL-1β ELISA.

## Author Contributions

**Conceptualization:** David Brough, Kevin N. Couper, Alister G. Craig.

**Data curation:** Janet Storm, Grazia Camarda.

**Formal analysis:** Janet Storm, Grazia Camarda.

**Funding acquisition:** David Brough, Kevin N. Couper, Alister G. Craig.

**Investigation:** Janet Storm, Grazia Camarda.

**Methodology:** Janet Storm, Grazia Camarda, Michael J. Haley, Kevin N. Couper, Alister G. Craig.

**Project administration:** Grazia Camarda.

**Supervision:** Alister G. Craig.

**Validation:** Janet Storm, Grazia Camarda.

**Visualization:** Janet Storm, Grazia Camarda.

**Writing – original draft:** Janet Storm.

**Writing – review & editing:** Grazia Camarda, Michael J. Haley, David Brough, Kevin N. Couper, Alister G. Craig.

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
