## [Decision Letter · Decision Letter 0]

14 Mar 2023

PONE-D-23-03391Plasmodium falciparum-infected erythrocyte co-culture with the monocyte cell line THP-1 does not trigger production of soluble factors reducing brain microvascular barrier functionPLOS ONE

Dear Dr. Storm,

Thank you for submitting your manuscript to PLOS ONE. After careful consideration, we feel that it has merit but does not fully meet PLOS ONE’s publication criteria as it currently stands. Therefore, we invite you to submit a revised version of the manuscript that addresses the points raised during the review process.

We look forward to receiving your revised manuscript.

Kind regards,

Gebreselassie Demeke

Academic Editor

PLOS ONE

Journal Requirements:

Reviewers' comments:

Reviewer's Responses to Questions

**Comments to the Author**

1. Is the manuscript technically sound, and do the data support the conclusions?

Reviewer #1: Yes

2. Has the statistical analysis been performed appropriately and rigorously? 

Reviewer #1: Yes

3. Have the authors made all data underlying the findings in their manuscript fully available?

Reviewer #1: Yes

4. Is the manuscript presented in an intelligible fashion and written in standard English?

Reviewer #1: Yes

5. Review Comments to the Author

Reviewer #1: The authors set out “to investigate Plasmodium falciparum-infected erythrocyte co-culture with the monocyte cell line THP-1 does not trigger production of soluble factors reducing brain microvascular barrier function”. They co-cultured IT4var14 IE and the monocyte cell line THP-1 for 24 hours and determined whether generated soluble molecules affect barrier function of human brain microvascular endothelial cells. They found that the medium produced after co-culture did not affect endothelial barrier function and similarly no effect was measured after inducing oxidative stress. Anyway, the manuscript can be improved in the following ways:

ABSTRACT

It is better, if it includes the study period,data entry, data collection tool and data analysis methods

Itroduction

Line 35: Plasmodium falciparum (P. falciparum), then use the abbreviated one throughout the whole document.

Line 41: Nk cells (please use the long fom at the first time), then use the abrivated one throughout the whole document.

Line 49: DNA(please write in long form at the 1st time)

Please do the rest in the same way

METHODS

It is better, if it includes the study period, data collection tool and study area (Where??? culture done)

Please, put the materials and methods after the introduction section not after the discussion section

RESULTS: It is ok

DISCUSSION: It is ok

SUMMARY: The article is an original and it gives an important clue. But, It needs minor revision.

6. PLOS authors have the option to publish the peer review history of their article (what does this mean?). If published, this will include your full peer review and any attached files.

Reviewer #1: No

---

## [Author Response · Author response to Decision Letter 0]

18 Apr 2023

We thank the reviewer for their comments and below is our point-by-point response to their comments and suggestions.

Abstract: It is better, if it includes the study period, data entry, data collection tool and data analysis methods.

 Data collection and analysis is written in the materials and methods section, and we have added a sentence at the start of that section with the study period. We do not think it is necessary to add this information to abstract. 

Introduction

Line 35: Plasmodium falciparum (P. falciparum), then use the abbreviated one throughout the whole document.

 We have written the full name (Plasmodium falciparum) in the abstract and the first time in the introduction and then shortened to P. falciparum. It is not common to have P. falciparum in brackets after the full name, it’s not really an abbreviation.

Line 41: Nk cells (please use the long form at the first time), then use the abbreviated one throughout the whole document.

 We now have written the full name (natural killer cells) in the 2 instances it appears in the manuscript.

Line 49: DNA (please write in long form at the 1st time). 

 We think this is not needed; it is a universal abbreviation.

Please do the rest in the same way 

 In addition to the abbreviations that were already defined in the manuscript, we added the full name, with the abbreviation in brackets, for the following: NLRP3, IL, PBMC, ICAM-1, LPS, SD and SEM. In a few instances we have removed the abbreviation and used the full name when it only appeared once or twice in the manuscript. These are: pathogen-associated molecular pattern, damage associated molecular patterns, glycosylphosphatidylinositol and interferon-gamma.

Methods

It is better, if it includes the study period, data collection tool and study area (Where??? culture done).

 We added a sentence at the start of the materials and methods section that the experiments were conducted at LSTM from January 2019 till October 2021. Data collection is described in each of the methods sections.

Please, put the materials and methods after the introduction section not after the discussion section. 

 We have moved the materials and methods section after the introduction and subsequently adjusted the appearance of the full name/abbreviation in order of the text.

RESULTS: It is ok

DISCUSSION: It is ok

SUMMARY: The article is an original and it gives an important clue. But, it needs minor revision.

 We are pleased that the reviewer finds our manuscript original and important and has no further requirements to revise our results and discussion sections.

---

## [Editor Report · Decision Letter 1]

20 Apr 2023

*Plasmodium falciparum*-infected erythrocyte co-culture with the monocyte cell line THP-1 does not trigger production of soluble factors reducing brain microvascular barrier function

PONE-D-23-03391R1

Dear Dr. Storm,

We’re pleased to inform you that your manuscript has been judged scientifically suitable for publication and will be formally accepted for publication once it meets all outstanding technical requirements.

Kind regards,

Gebreselassie Demeke

Academic Editor
---

## [Editor Report · Acceptance letter]

26 Apr 2023

PONE-D-23-03391R1 

*Plasmodium falciparum*-infected erythrocyte co-culture with the monocyte cell line THP-1 does not trigger production of soluble factors reducing brain microvascular barrier function 

Dear Dr. Storm:

I'm pleased to inform you that your manuscript has been deemed suitable for publication in PLOS ONE. Congratulations! Your manuscript is now with our production department. 

Kind regards, 

on behalf of

Dr. Gebreselassie Demeke 

Academic Editor

PLOS ONE